# Opioid-related overdose and chronic use following an initial prescription of hydrocodone versus oxycodone

Scott G. Weiner[1,2]☯*, Michelle A. Hendricks[3]☯, Sanae El Ibrahimi[3,4]☯, Grant A. Ritter[5], Sara E. Hallvik[3], Christi Hildebran[3], Roger D. Weiss[2,6], Edward W. Boyer[7], Diana P. Flores[3], Lewis S. Nelson[8], Peter W. Kreiner[5], Michael A. Fischer[9]

**1** Department of Emergency Medicine, Brigham and Women's Hospital, Boston, Massachusetts, United States of America, **2** Harvard Medical School, Boston, Massachusetts, United States of America, **3** Division of Research and Evaluation, Comagine Health, Portland, Oregon, United States of America, **4** Department of Epidemiology and Biostatistics, School of Public Health, University of Nevada, Las Vegas, Las Vegas, Nevada, United States of America, **5** Brandeis University, Waltham, Massachusetts, United States of America, **6** Department of Psychiatry, McLean Hospital, Belmont, Massachusetts, United States of America, **7** Department of Emergency Medicine, Ohio State University, Columbus, Ohio, United States of America, **8** Department of Emergency Medicine Rutgers New Jersey Medical School, Newark, New Jersey, United States of America, **9** Section of General Internal Medicine, Boston Medical Center, Boston University School of Medicine, Boston, Massachusetts, United States of America

☯ These authors contributed equally to this work.
* sweiner@bwh.harvard.edu

**Data Availability Statement:** Data cannot be shared publicly because of data use limitations. The data underlying the results presented in the study are available from the Oregon Health

## Abstract

### Background

Hydrocodone and oxycodone are prescribed commonly to treat pain. However, differences in risk of opioid-related adverse outcomes after an initial prescription are unknown.

This study aims to determine the risk of opioid-related adverse events, defined as either chronic use or opioid overdose, following a first prescription of hydrocodone or oxycodone to opioid naïve patients.

### Methods

A retrospective analysis of multiple linked public health datasets in the state of Oregon. Adult patients ages 18 and older who a) received an initial prescription for oxycodone or hydrocodone between 2015–2017 and b) had no opioid prescriptions or opioid-related hospitalizations or emergency department visits in the year preceding the prescription were followed through the end of 2018. First-year chronic opioid use was defined as ≥6 opioid prescriptions (including index) and average ≤30 days uncovered between prescriptions. Fatal or non-fatal opioid overdose was indicated from insurance claims, hospital discharge data or vital records.

### Results

After index prescription, 2.8% (n = 14,458) of individuals developed chronic use and 0.3% (n = 1,480) experienced overdose. After adjustment for patient and index prescription

Authority Public Health Division at phone number 971-673-1222 or email health. webmaster@dhsoha.state.or.us.

**Funding:** National Institute on Drug Abuse 5-R01-DA044167 (SGW, MAH, SEI, GAR, SEH, CH, RDW, EWB, DPF, PWK) and Agency for Healthcare Research and Quality 5-R01-HS026753 (SGW). The funders had no role in study design, data collection and analysis, decision to publish, or preparation of the manuscript.

**Competing interests:** The authors have declared that no competing interests exist.

characteristics, patients receiving oxycodone had lower odds of developing chronic use relative to patients receiving hydrocodone (adjusted odds ratio = 0.95, 95% confidence interval (CI) 0.91–1.00) but a higher risk of overdose (adjusted hazard ratio (aHR) = 1.65, 95% CI 1.45–1.87). Oxycodone monotherapy appears to greatly increase the hazard of opioid overdose (aHR 2.18, 95% CI 1.86–2.57) compared with hydrocodone with acetaminophen. Oxycodone combined with acetaminophen also shows a significant increase (aHR 1.26, 95% CI 1.06–1.50), but not to the same extent.

## Conclusions

Among previously opioid-naïve patients, the risk of developing chronic use was slightly higher with hydrocodone, whereas the risk of overdose was higher after oxycodone, in combination with acetaminophen or monotherapy. With a goal of reducing overdose-related deaths, hydrocodone may be the favorable agent.

## Introduction

Hydrocodone or oxycodone? When prescribing opioids for pain, prescribers in the U.S. commonly decide between these two medications. Although the total number of opioid prescriptions has declined in recent years, hydrocodone remains the most frequently prescribed opioid with 83.6 million prescriptions dispensed in 2017 [1,2]. Oxycodone is prescribed less frequently (55.2 million in 2017), but still represents roughly half of the opioids consumed in the U.S. when calculated as total morphine milligram equivalents (MMEs) [3,4]. Although national data on opioid prescriptions to opioid-naïve patients is limited, our prior study based on the Ohio prescription drug monitoring program determined that 50.8% of prescriptions to opioid-naïve patients were for hydrocodone and 25.1% were for oxycodone [5]. A national sample of 19 emergency departments (EDs) discovered that the most common opioids prescribed in that setting were oxycodone (52.3%) and hydrocodone (40.9%) [6].

Both oxycodone and hydrocodone are indicated for treatment of moderate to severe pain. For acute pain, the two drugs have similar efficacy [7–9]. Since the drugs have the same mechanism, the question remains if their risk profiles differ. Hydrocodone was once considered a safer opioid, classified as a Drug Enforcement Administration schedule III drug, similar to some formulations of codeine and other less potent opioids. However, hydrocodone was reclassified in 2014 to schedule II, the category containing oxycodone; schedule II is reserved for prescribed medications with the highest misuse potential [10,11]. Despite being in the same category, it is unclear if the two drugs are similar in their potential for misuse. Oxycodone is theorized to have greater misuse liability due to its higher likability scores and a lack of negative subjective effects when compared to hydrocodone and morphine [12]. The Ohio study determined that, compared to codeine, first-time prescriptions of hydrocodone were associated with an odds ratio of 1.33 (95% confidence interval (CI) 1.31–1.35) for developing chronic use, and those of oxycodone had an odds ratio of 1.64 (95% CI 1.61–1.67) [5]. Another study of a sample of commercially insured patients determined that probability of long-term use after a first prescription was 5.1% for short-acting hydrocodone and 4.7% for short-acting oxycodone [13]. Neither of these studies controlled for dose, duration of therapy or patient level factors. In other studies of people who misuse drugs and in opioid naïve patients,

however, no differences were seen in the pharmacodynamic effects or misuse liability of either drug [14,15].

Chronic opioid use is a negative outcome given the risks of chronic opioid therapy, including development of dependence and other medical complications such as myocardial infarction and fractures [16]. Many of the risks are dose dependent, increasing with higher morphine milligram equivalent use [17]. Furthermore, tapering from stable but higher dose (≥50 MME/day) chronic opioid therapy is associated with increased risk of emergency department or hospital encounters for drug overdose or withdrawal and mental health crisis [18]. Due to these and other complications, avoidance of chronic opioid therapy for pain not related to malignancy is recommended by the Centers for Disease Control and Prevention's 2016 opioid guidelines, although an updated version is currently forthcoming [19,20].

Potentially the most dangerous outcome after initiation of opioid therapy is overdose, either of prescribed opioids or after transition to non-prescribed opioids, such as heroin or illicitly manufactured fentanyl. In our prior work, we found that 3 out of every 1,000 opioid naïve patients prescribed an opioid had a subsequent opioid overdose, and the risks were higher for certain groups, such as the elderly and those with multiple medical comorbidities or certain psychiatric comorbidities [21]. In Oregon, the state where this study took place, at least 300 people have died from opioid-related overdose each year since 2015 [22]. Deaths associated with prescribed opioids have remained stable, but there has been a rapid increase in deaths from heroin and synthetic opioids since the start of the COVID-19 pandemic [23].

Although current rising trends are not driven by prescriptions, there is still a need to ensure safe prescription opioid practices. Because oxycodone and hydrocodone appear to be used interchangeably for acute pain, this research aims to determine if there is variability in risk-related outcomes between patients given a first prescription of oxycodone versus hydrocodone. Using a merged public health database from Oregon, we compared the rates of chronic use or opioid overdose after an opioid-naïve patient is prescribed either of these medications. Because oxycodone, but not hydrocodone, is commonly prescribed in two different formulations, we also compared risks for oxycodone prescribed alone or as a combination pill along with acetaminophen. We also evaluated the cumulative MME that patients received in the first 6 months following a first prescription and the rate of switching between different opioids after the first prescription as secondary outcomes.

## Methods

### Study design and data sources

This was a retrospective cohort study using the Oregon Comprehensive Opioid Risk Registry, a dataset formed by merging several previously disparate public health datasets in that state and described in detail elsewhere [24]. In brief, 2014–2018 patient claims data from the Voluntary Oregon All Payer All Claims Database (APCD) were linked to several Oregon public health datasets including the Prescription Drug Monitoring Program (PDMP), Hospital Discharge Database (HDD), and Vital Records. Using the R package "fastLink", datasets were probabilistically linked within the Oregon Health Authority Public Health Division using first name, last name, and date of birth [25]. The terms of the data use agreement prohibit public sharing of the source data used in this study but may be obtained via request to the public health entities in Oregon that oversee the individual source datasets.

The Oregon APCD includes claims for all medical services and prescriptions paid for by participating health plans. The dataset includes 100% of the Medicaid population, 85% of the Medicare Advantage population, 81% of the fully insured commercial population, and 24%

of the self-insured commercial population, in total representing approximately 80% of all insured Oregonians. In the U.S., Medicaid is a state-run safety net insurance for low-income individuals of any age, Medicare Advantage is a federal government-administered insurance program primarily for individuals ≥65 years old (as well as younger disabled people and dialysis patients) which is managed by a commercial insurance company, commercial insurance is a privately managed plan generally serving employed individuals and their families, and dual Medicaid/Medicare Advantage enrollees qualify for both plans, typically due to a combination of age, disability, medical comorbidities and low income. Patients in the Medicare Fee-for-Service population, which is the traditional public insurance plan for individuals ≥65 years old, were not included due to data availability limitations. The APCD served as the primary source for patient characteristics and opioid-related ED visits. The PDMP, HDD, and Vital Records datasets provided data on dispensed opioid prescriptions, opioid-related inpatient hospitalizations, and opioid poisoning deaths for all Oregon residents, regardless of payer.

**Inclusion criteria.** The sample included opioid-naïve adults who received a first prescription (termed the index prescription) of short-acting hydrocodone or short-acting oxycodone between 2015–2017. To be considered opioid-naïve, patients must not have had any opioid prescriptions, opioid-related inpatient hospitalizations, or opioid-related ED visits in the 12 months preceding the index prescription. Patients were excluded if they did not have continuous insurance enrollment for the year that they received the index prescription (gaps less than 90 days were allowed). Patients who would have otherwise met continuous enrollment criteria but died were not excluded.

**Patient characteristics.** Patient characteristics originated in the APCD and included age, race, ethnicity, sex, insurance type, and urbanization of the county of residence. Age was defined as of January 1, 2014 and divided into 7 categories. Because the Oregon APCD enrollment files are missing race and ethnicity for about half of patients, the Bayesian Imputed Surname Geocoding (BISG) algorithm was used to impute missing race and ethnicity when adequate information was available to do so [26]. The imputation used the following categories: White, Black, Hispanic, Asian-Pacific Islander, and other. American Indian and Alaska Native race was not imputed due to inadequate reliability for BISG estimation in this race group [27]. Imputation was implemented using the R package "wru" [28].

The type of insurance, either Medicaid, Medicare Advantage, commercial, or dual Medicaid/Medicare Advantage, was based on the insurance that the patient was enrolled in for at least 6 months of the calendar year in which they received the index prescription. The 2013 National Center for Health Statistics urban-rural classification scheme was used to classify urbanization based on patients' county of residence for the year they received the index prescription: large central metro, large fringe metro, medium metro, small metro, micropolitan, and noncore counties [29].

**Prescription characteristics.** Food and Drug Administration National Drug Codes were used to identify hydrocodone, oxycodone, and other opioids in the PDMP dataset based on pharmaceutical class and drug names [30]. The PDMP data were cleaned to remove duplicate entries, those missing a prescriber ID, quantities less than 4 or greater than or equal to the 99th percentile (n = 268), and days' supply less than zero or greater than 90 days, similar to prior work [5]. Long-acting formulations were excluded due to the small percentage of index prescriptions that were long-acting in our sample (e.g., 0.05% of hydrocodone or oxycodone index prescriptions in 2015). The total morphine milligram equivalent (MME) for the index prescription was calculated using Centers for Disease Control and Prevention conversion reference tables [31]. MME was placed into 5 categories based on the following sample percentiles: 25th, 50th, 75th, and 90th. Days' supply, as provided to the PDMP by the pharmacy at time

of prescription fill, was categorized as follows: 3 or less days' supply, 4–6 days' supply, 7+ days' supply.

## Primary outcomes

**First-year chronic opioid use.**   Chronic use was defined similarly to a previously published definition [5]. In brief, the following criteria had to have been met in the 365 days after the index prescription: a) the patient must have filled six prescriptions of any opioid type, including the index prescription, and b) periods uncovered by an opioid prescription, on average, must have been 30 days or less. The uncovered period between refills was calculated by taking the difference between the predicted date of the refill based on days' supply and the actual refill date. For the last prescription of the year, the date 365 days from the index prescription was used instead of a refill date to calculate uncovered days. The average period uncovered by an opioid prescription (criteria b) was calculated by dividing the total uncovered days by the number of gaps between refills plus one (to reflect the uncovered gap from the last prescription of the year to the end of the year).

To test our criteria, we also calculated an alternative definition of chronic use that defines long-term use as episodes exceeding 90 days with either a total days' supply greater than or equal to 120 or 10+ prescriptions filled [32]. For this analysis, the onset of the index prescription was considered the onset of the episode, and the last date of an opioid fill in the 365 days after the index prescription was considered the end of the episode. Also, our chronic use definition did not exclude people who died within a year of the initial dispensation, so not all individuals had the same follow-up period to be classified as chronic users. Therefore, we performed an additional sensitivity analysis that excluded individuals who died in the year following their index prescription.

**Opioid overdose.**   This outcome was the time from the index prescription to the first incident fatal or non-fatal opioid overdose from either illicit or prescription opioids. Survival days were calculated as the number of days from the index prescription to either: loss of continuous insurance enrollment, the first nonfatal or fatal opioid overdose, death from a non-opioid-related cause, or the end of the study follow-up period on December 31, 2018. To meet data use agreement restrictions limiting use of enrollment dates, continuous enrollment was determined in one-year periods, so loss of continuous enrollment at any time during a calendar year, after the index year, resulted in a censor date on December 31 of the preceding year. Fatal opioid overdoses were identified from Vital Records using ICD-10 underlying cause of death X40-X44, X60-X64, X85, Y10-Y14 with multiple cause-of-death codes (T40.0, T40.1, T40.2, T40.3, T40.4, T40.6), following guidance from the Substance Abuse and Mental Health Services Administration [33]. Literal text fields in the death record were also searched for terms associated with opioid overdose deaths using a tool developed by the Council of State and Territorial Epidemiologists Overdose Subcommittee [34]. Nonfatal opioid overdoses were derived from diagnosis codes in hospital discharge records and ED claims in the APCD (see S1 Table in the supplemental materials for the list of codes used).

## Secondary outcomes

**Cumulative MME.**   To explore whether the drug type of the initial index prescription is associated with subsequent dosages, we calculated a cumulative MME for all opioid fills in the 6 months after the index prescription, excluding the index prescription.

**Drug switching.**   Drug switching was defined as a replacement of the index drug (oxycodone or hydrocodone) with any other opioid within the first six months after the initial index

prescription was filled. We created a flag that indicated if the patient switched drugs in the follow-up period.

## Statistical analyses

We computed frequencies for patient characteristics (age, gender, ethnicity/race, insurance type in the index year, urbanization in county of residence in the index year) and prescription characteristics MME category, days' supply) for the full cohort and by index prescription drug. Standardized differences between index drugs were calculated using Cohen's *d* effect size index [35]. To obtain adjusted odds ratios for first-year chronic use and drug switching, we conducted multilevel logistic regression models, including fixed effects for patient and index prescription characteristics and a random effect for prescriber. We also conducted a sensitivity analysis to examine findings using an alternative, commonly used chronic use definition. Cox proportional hazard models were used to calculate adjusted hazard ratios for the risk of overdose during the follow-up period. Computer memory limitations did not allow us to add a random effect for prescriber to the Cox models for the full sample, so we conducted a subgroup analysis on the subset of prescribers with prescriptions written to at least 200 patients. A zero-inflated negative binomial model assessed the probability of having zero cumulative MME in the first 6 months after the index prescription (logistic portion) and predictors of the 6-month cumulative MME (count portion), conditioned on being non-zero. Analyses were conducted in SAS version 9.4 (SAS Institute Inc., Cary, NC, USA).

**Combination medication subgroup analysis.** In our data, prescriptions of immediate-release hydrocodone were always combined with other active ingredients (e.g., acetaminophen and ibuprofen), whereas oxycodone was equally commonly prescribed either alone or in combination with other the other active ingredients. Given this difference in typical formulations for oxycodone or hydrocodone, we could not be certain whether our findings were reflective of differing opioid risk profiles or due to differences in risk for oxycodone-acetaminophen combination therapy compared to opioid monotherapy. Previous studies comparing oxycodone-acetaminophen combination therapy with oxycodone monotherapy suggest that combination therapy provides similar pain relief with fewer side effects [36,37]. Oxycodone combined with acetaminophen may have a synergistic mechanism of action leading to overall decreased doses to treat moderate or severe pain, which may reduce the risk for opioid-related adverse events such as chronic use or overdose [38]. To examine this possibility, we conducted a sub-analysis of individuals who received oxycodone monotherapy, a combination oxycodone-acetaminophen index drug, or a combination hydrocodone-acetaminophen index drug with strength-per-unit of 5 mg.

**Compliance.** This was a retrospective study of administrative public health datasets. After dataset linkage was performed on identifiable data, all identifiers were destroyed and subsequent data analysis was performed on strictly deidentified data. Study activities were approved by the Mass General Brigham Human Research Committee, which waived the requirement for informed consent as the research was minimal risk and could not reasonably be carried out without the waiver. The Oregon Health Authority Public Health Division ceded oversight to the approving entity.

## Results

### Patients

Characteristics of the 519,066 opioid-naïve patients who met the inclusion criteria are shown in Table 1 (see S1 Fig in the supplemental materials for a participant flow diagram). Most were between the ages of 25 and 64 (72.7%), female (55.2%), White (73.5%) and lived in either a

**Table 1. Patient and index prescription characteristics, by index prescription drug type.**

| | Full Cohort | | Hydrocodone SA Treatment | | Oxycodone SA Treatment | | Standardized Differences |
|---|---|---|---|---|---|---|---|
| | N | % | N | % | N | % | |
| **Age** | | | | | | | |
| 18–24 | 66,110 | 12.74 | 46,361 | 13.14 | 19,749 | 11.89 | 0.038 |
| 25–34 | 106,369 | 20.49 | 70,898 | 20.09 | 35,471 | 21.35 | 0.031 |
| 35–44 | 90,574 | 17.45 | 61,222 | 17.35 | 29,352 | 17.67 | 0.009 |
| 45–54 | 90,152 | 17.37 | 61,827 | 17.52 | 28,325 | 17.05 | 0.012 |
| 55–64 | 90,450 | 17.43 | 61,304 | 17.37 | 29,146 | 17.55 | 0.005 |
| 65–74 | 48,212 | 9.29 | 32,150 | 9.11 | 16,062 | 9.67 | 0.019 |
| 75+ | 27,199 | 5.24 | 19,191 | 5.44 | 8,008 | 4.82 | 0.028 |
| **Gender** | | | | | | | |
| F | 286,488 | 55.19 | 190,687 | 54.03 | 95,801 | 57.67 | 0.073 |
| M | 232,578 | 44.81 | 162,266 | 45.97 | 70,312 | 42.33 | 0.073 |
| **Race/Ethnicity** | | | | | | | |
| White | 381,580 | 73.51 | 258,135 | 73.14 | 123,445 | 74.31 | 0.027 |
| Black | 16,383 | 3.16 | 10,910 | 3.09 | 5,473 | 3.29 | 0.012 |
| Hispanic | 47,350 | 9.12 | 33,323 | 9.44 | 14,027 | 8.44 | 0.035 |
| Asian-Pacific Islander | 12,523 | 2.41 | 8,564 | 2.43 | 3,959 | 2.38 | 0.003 |
| Other | 7,908 | 1.52 | 5,598 | 1.59 | 2,310 | 1.39 | 0.016 |
| Unknown | 53,322 | 10.27 | 36,423 | 10.32 | 16,899 | 10.17 | 0.005 |
| **Insurance Plan in Index Year** | | | | | | | |
| Commercial | 232,408 | 44.77 | 155,868 | 44.16 | 76,540 | 46.08 | 0.039 |
| Medicaid | 196,972 | 37.95 | 136,672 | 38.72 | 60,300 | 36.30 | 0.050 |
| Medicare | 67,031 | 12.91 | 45,421 | 12.87 | 21,610 | 13.01 | 0.004 |
| Dual | 22,453 | 4.33 | 14,854 | 4.21 | 7,599 | 4.57 | 0.018 |
| Unknown | 202 | 0.04 | 138 | 0.04 | 64 | 0.04 | 0.000 |
| **Urbanization in Index Year** | | | | | | | |
| Large central metro | 93,689 | 18.05 | 60,065 | 17.02 | 33,624 | 20.24 | 0.083 |
| Large fringe metro | 114,626 | 22.08 | 73,610 | 20.86 | 41,016 | 24.69 | 0.092 |
| Medium metro | 98,412 | 18.96 | 68,742 | 19.48 | 29,670 | 17.86 | 0.041 |
| Small metro | 73,619 | 14.18 | 52,546 | 14.89 | 21,073 | 12.69 | 0.064 |
| Micropolitan | 48,618 | 9.37 | 35,776 | 10.14 | 12,842 | 7.73 | 0.084 |
| Noncore | 7,379 | 1.42 | 5,385 | 1.53 | 1,994 | 1.20 | 0.028 |
| Unknown | 82,723 | 15.94 | 56,829 | 16.10 | 25,894 | 15.59 | 0.014 |
| **Year of Index Prescription** | | | | | | | |
| 2015 | 202,070 | 38.93 | 140,357 | 39.77 | 61,713 | 37.15 | 0.054 |
| 2016 | 177,365 | 34.17 | 120,426 | 34.12 | 56,939 | 34.28 | 0.003 |
| 2017 | 139,631 | 26.90 | 92,170 | 26.11 | 47,461 | 28.57 | 0.055 |
| **Index Prescription MME** | | | | | | | |
| MME < = 75 | 152,349 | 29.35 | 138,274 | 39.18 | 14,075 | 8.47 | 0.773 |
| MME 76–100 | 108,985 | 21.00 | 100,861 | 28.58 | 8,124 | 4.89 | 0.669 |
| MME 101–200 | 127,982 | 24.66 | 72,381 | 20.51 | 55,601 | 33.47 | 0.295 |
| MME 201–300 | 85,245 | 16.42 | 31,293 | 8.87 | 53,952 | 32.48 | 0.610 |
| MME >300 | 44,505 | 8.57 | 10,144 | 2.87 | 34,361 | 20.69 | 0.575 |
| **Index Prescription Days' Supply** | | | | | | | |
| < = 3 Days | 292,031 | 56.26 | 213,278 | 60.43 | 78,753 | 47.41 | 0.263 |
| 4–6 Days | 151,032 | 29.10 | 92,024 | 26.07 | 59,008 | 35.52 | 0.206 |
| 7+ Days | 76,003 | 14.64 | 47,651 | 13.50 | 28,352 | 17.07 | 0.099 |

SA, short-acting; MME, morphine milligram equivalents.

large or medium-sized metro area (59.1%). The most common insurance type was commercial (44.8%). Standardized differences for patient characteristics between those receiving hydrocodone or oxycodone treatment were generally small or close to zero.

## Index prescription characteristics

A total of 18,758 unique prescribers prescribed index opioid prescriptions of hydrocodone or oxycodone, with a median of 8 patients per prescriber (interquartile range (IQR) = 2–29). About two-thirds (68.0%) of patients filled an index prescription of hydrocodone and 32.0% filled oxycodone. The majority of patients (56.3%) filled a days' supply less than or equal to 3 days. Standardized differences in dosages between patients receiving hydrocodone and those receiving oxycodone were large (Table 1). Most individuals who filled oxycodone received more than 100 MME (86.6%), while only 32.3% of those filling hydrocodone received more than 100 MME. Patients given oxycodone also received more days' supply, with 52.6% of patients receiving a days' supply greater than 3, compared to 39.6% of those with hydrocodone.

## Primary outcomes

**First-year chronic use.**    In the year after receiving the index prescription, 2.8% (n = 14,458) of patients met the criteria for chronic use. The median total days' supply for individuals meeting the chronic use definition was 190 (IQR = 115–300), compared to 5 (IQR = 3–10) for people not meeting the definition. The raw percentage of patients receiving oxycodone that developed chronic use was 3.3%, compared to 2.5% of those with hydrocodone. After adjustment for patient and index prescription characteristics, however, patients receiving an index prescription of oxycodone had *lower odds* of developing chronic use relative to patients receiving hydrocodone (aOR = 0.95, 95% CI 0.91–1.00; Table 2). Further exploration of model results indicated that the change in relationship between drug type and chronic use after controlling for patient and index prescription characteristics was due to the addition of dosage to the model. When index prescription MME is excluded from the model, patients filling an oxycodone index prescription had significantly *higher odds* of developing chronic use relative to patients receiving hydrocodone (aOR = 1.36, 95% CI 1.29–1.42).

The sensitivity analysis, using an alternative version of the chronic use definition, identified slightly more patients who meet criteria for chronic use (3.1%; n = 16,297). The median total days' supply for people meeting the alternative chronic use definition was 174 (IQR = 116–282). There was significant overlap between the definitions: 85.3% of those who met the alternative definition also met our study definition, while 96.2% of those who met our study definition also met the alternative definition. S2 Table in the supplemental materials shows odds of first-year chronic use by index prescription and patient characteristics using the alternative definition. Consistent with the analyses using our study definition, individuals receiving an oxycodone index prescription had lower odds of chronic use (aOR = 0.91, 95% CI 0.87–0.95). Likewise, to address not including individuals who died in the year following their index prescription, we ascertained that 1.1% (n = 3,750) patients who received an index prescription of hydrocodone SA died in the following year compared with 1.7% (n = 2,868) who received oxycodone SA. Excluding the individuals who died during the follow-up year did not substantially change the chronic user results (oxycodone OR 0.94 (95% CI 0.89–0.98) vs. OR 0.95 (95% CI 0.91–1.0) compared with hydrocodone).

**Opioid overdose.**    Opioid overdose was experienced by 0.3% (n = 1,480) of patients prior to censoring. The percentage of patients who filled oxycodone experiencing overdose was 0.38%, compared to 0.24% of those receiving hydrocodone. Table 3 shows adjusted hazard

**Table 2. Patient and index prescription characteristics associated with first-year chronic use.**

| | No Chronic Use (n) | Chronic Use (n) | Chronic Use Row % | Adjusted Odds Ratio (95% Confidence Interval) | p value |
|---|---|---|---|---|---|
| **Total Cohort** | 504,608 | 14,458 | 2.79 | | |
| **Index Prescription Drug** | | | | | |
| Hydrocodone SA | 344,029 | 8,924 | 2.53 | ref | ref |
| Oxycodone SA | 160,579 | 5,534 | 3.33 | 0.95 (0.91–1.00) | 0.03 |
| **Age** | | | | | |
| 18–24 | 65,558 | 552 | 0.83 | Ref | ref |
| 25–34 | 104,571 | 1,798 | 1.69 | 1.81 (1.64–2.00) | <.0001 |
| 35–44 | 88,277 | 2,297 | 2.54 | 2.71 (2.46–2.98) | <.0001 |
| 45–54 | 86,966 | 3,186 | 3.53 | 3.57 (3.25–3.92) | <.0001 |
| 55–64 | 86,932 | 3,518 | 3.89 | 3.82 (3.47–4.20) | <.0001 |
| 65–74 | 46,433 | 1,779 | 3.69 | 2.96 (2.62–3.33) | <.0001 |
| 75+ | 25,871 | 1,328 | 4.88 | 3.06 (2.70–3.47) | <.0001 |
| **Gender** | | | | | |
| F | 278,826 | 7,662 | 2.67 | 0.98 (0.95–1.02) | 0.27 |
| M | 225,782 | 6,796 | 2.92 | ref | ref |
| **Race/Ethnicity** | | | | | |
| White | 370,164 | 11,416 | 2.99 | ref | ref |
| Black | 15,847 | 536 | 3.27 | 1.16 (1.05–1.28) | 0.00 |
| Hispanic | 46,515 | 835 | 1.76 | 0.60 (0.56–0.65) | <.0001 |
| Asian-Pacific Islander | 12,393 | 130 | 1.04 | 0.37 (0.31–0.44) | <.0001 |
| Other | 7,603 | 305 | 3.86 | 1.09 (0.96–1.24) | 0.17 |
| Unknown | 52,086 | 1,236 | 2.32 | 0.90 (0.84–0.97) | 0.005 |
| **Insurance Plan in Index Year** | | | | | |
| Commercial | 228,823 | 3,585 | 1.54 | ref | ref |
| Medicaid | 190,262 | 6,710 | 3.41 | 2.86 (2.73–3.00) | <.0001 |
| Medicare | 64,750 | 2,281 | 3.40 | 1.56 (1.43–1.69) | <.0001 |
| Dual | 20,575 | 1,878 | 8.36 | 3.75 (3.48–4.04) | <.0001 |
| Unknown | 198 | 4 | 1.98 | 1.36 (0.47–3.91) | 0.57 |
| **Urbanization in Index Year** | | | | | |
| Large central metro | 91,300 | 2,389 | 2.55 | 0.88 (0.75–1.03) | 0.11 |
| Large fringe metro | 111,842 | 2,784 | 2.43 | 0.94 (0.80–1.10) | 0.43 |
| Medium metro | 95,858 | 2,554 | 2.60 | 0.92 (0.79–1.08) | 0.30 |
| Small metro | 71,519 | 2,100 | 2.85 | 0.92 (0.79–1.08) | 0.30 |
| Micropolitan | 46,976 | 1,642 | 3.38 | 0.98 (0.84–1.15) | 0.81 |
| Noncore | 7,118 | 261 | 3.54 | ref | ref |
| Unknown | 79,995 | 2,728 | 3.30 | 1.02 (0.88–1.19) | 0.78 |
| **Year of Index Prescription** | | | | | |
| 2015 | 195,521 | 6,549 | 3.24 | ref | ref |
| 2016 | 172,987 | 4,378 | 2.47 | 0.84 (0.81–0.88) | <.0001 |
| 2017 | 136,100 | 3,531 | 2.53 | 0.88 (0.84–0.92) | <.0001 |
| **Index Prescription MME** | | | | | |
| MME <= 75 | 149,757 | 2,592 | 1.70 | ref | ref |
| MME 76–100 | 107,207 | 1,778 | 1.63 | 1.03 (0.96–1.10) | 0.39 |
| MME 101–200 | 124,579 | 3,403 | 2.66 | 1.48 (1.38–1.58) | <.0001 |
| MME 201–300 | 82,382 | 2,863 | 3.36 | 1.84 (1.70–1.99) | <.0001 |
| MME >300 | 40,683 | 3,822 | 8.59 | 3.89 (3.57–4.24) | <.0001 |
| **Index Prescription Days' Supply** | | | | | |

*(Continued)*

**Table 2.** (Continued)

| | No Chronic Use (n) | Chronic Use (n) | Chronic Use Row % | Adjusted Odds Ratio (95% Confidence Interval) | p value |
|---|---|---|---|---|---|
| < = 3 Days | 287,321 | 4,710 | 1.61 | ref | ref |
| 4–6 Days | 147,755 | 3,277 | 2.17 | 1.08 (1.03–1.14) | 0.004 |
| 7+ Days | 69,532 | 6,471 | 8.51 | 2.49 (2.34–2.64) | < .0001 |

SA, short-acting; MME, morphine milligram equivalents.

ratios for index prescription characteristics associated with subsequent opioid overdose, adjusting for patient and prescription characteristics. The adjusted hazard ratio (aHR) of having an overdose for individuals filling an oxycodone index prescription was 1.65 (95% CI 1.45–1.87) relative to those filling hydrocodone. To account for random variation among prescribers, we repeated the analysis above, but included a random prescriber effect on a subgroup of patients from prescribers with at least 200 patients who filled an index prescription of hydrocodone or oxycodone. The sample included 101,559 patients and 294 prescribers. Patients in the sub-sample experienced 287 overdoses during the study period. Results of this sub-analysis were similar to the overall sample findings, with a greater hazard of overdose for individuals filling an index prescription of oxycodone (aHR = 1.75, 95% CI 1.29–2.37) (S3 Table).

## Secondary outcomes

**Cumulative MME.** After controlling for patient and prescription characteristics, patients receiving oxycodone were less likely than those with hydrocodone to have zero cumulative MME in the 6 months after the index prescription (aOR = 0.88, 95% CI 0.87–0.89), meaning that patients with oxycodone were more likely to have filled at least one additional opioid prescription after the index prescription fill. Cumulative 6-month MME for patients receiving oxycodone was 1.20 times greater than for those with hydrocodone (95% CI 1.18–1.22). The mean cumulative MME for patients with oxycodone was 393.85 (standard deviation (SD) = 2,013.34) and for those with hydrocodone was 213.37 (SD = 1,151.75).

**Drug switching.** Among the 215,726 patients with at least one additional prescription in the 6 months after the index prescription (representing 41.6% of the total cohort), more than a third (37.8%, n = 81,495) switched to another opioid. Of those patients who started on oxycodone, 41.5% switched to another opioid within 6 months (n = 29,977), while 35.9% of those started on hydrocodone switched (n = 51,518). After adjusting for patient and index prescription characteristics, patients whose index prescription was oxycodone were 1.24 times more likely to be switched to another opioid than those receiving hydrocodone (95% CI 1.21–1.27).

## Combination product subgroup analysis

There were 478,133 patients in the sample who filled an index prescription for 5 mg strength immediate release oxycodone or hydrocodone. Of these, 319,942 filled hydrocodone-acetaminophen therapy (66.9%), 75,810 filled oxycodone-acetaminophen therapy (15.9%), and 82,381 filled oxycodone monotherapy (17.2%).

First-year chronic use was experienced by 2.5% (n = 12,004) of individuals in this subgroup analysis. Of those who filled an oxycodone monotherapy index prescription, 3.5%

**Table 3. Patient and index prescription characteristics associated with opioid overdose (fatal or non-fatal).**

| | No Overdose | Overdose | Overdose Row % | Adjusted Hazard Ratio (95% Confidence Interval) | p value |
|---|---|---|---|---|---|
| **Total Cohort** | 517,586 | 1,480 | 0.29 | | |
| **Index Prescription Drug** | | | | | |
| Hydrocodone SA | 352,096 | 857 | 0.24 | ref | ref |
| Oxycodone SA | 165,490 | 623 | 0.38 | 1.65 (1.45–1.87) | <0.01 |
| **Age** | | | | | |
| 18–24 | 65,862 | 248 | 0.38 | ref | ref |
| 25–34 | 106,069 | 300 | 0.28 | 0.71 (0.60–0.84) | < .0001 |
| 35–44 | 90,358 | 216 | 0.24 | 0.66 (0.55–0.80) | < .0001 |
| 45–54 | 89,909 | 243 | 0.27 | 0.76 (0.64–0.91) | 0.00 |
| 55–64 | 90,235 | 215 | 0.24 | 0.77 (0.64–0.93) | 0.01 |
| 65–74 | 48,087 | 125 | 0.26 | 0.76 (0.56–1.03) | 0.08 |
| 75+ | 27,066 | 133 | 0.49 | 1.45 (1.08–1.96) | 0.02 |
| **Gender** | | | | | |
| F | 285,770 | 718 | 0.25 | 0.68 (0.62–0.76) | <0.01 |
| M | 231,816 | 762 | 0.33 | ref | ref |
| **Race/Ethnicity** | | | | | |
| White | 380,402 | 1,178 | 0.31 | ref | ref |
| Black | 16,312 | 71 | 0.43 | 0.96 (0.75–1.23) | 0.76 |
| Hispanic | 47,258 | 92 | 0.19 | 0.50 (0.40–0.62) | <0.01 |
| Asian-Pacific Islander | 12,512 | 11 | 0.09 | 0.24 (0.13–0.43) | <0.01 |
| Other | 7,867 | 41 | 0.52 | 1.13 (0.83–1.55) | 0.44 |
| Unknown | 53,235 | 87 | 0.16 | 0.93 (0.74–1.18) | 0.57 |
| **Insurance Plan in Index Year** | | | | | |
| Commercial | 232,197 | 211 | 0.09 | ref | ref |
| Medicaid | 196,050 | 922 | 0.47 | 5.39 (4.61–6.30) | <0.01 |
| Medicare | 66,855 | 176 | 0.26 | 2.12 (1.58–2.84) | <0.01 |
| Dual | 22,283 | 170 | 0.76 | 7.15 (5.65–9.04) | <0.01 |
| Unknown | 201 | 1 | 0.50 | 5.38 (0.75–38.45) | 0.09 |
| **Urbanization in Index Year** | | | | | |
| Large central metro | 93,337 | 352 | 0.38 | ref | ref |
| Large fringe metro | 114,361 | 265 | 0.23 | 0.74 (0.63–0.87) | <0.01 |
| Medium metro | 98,115 | 297 | 0.30 | 0.83 (0.71–0.98) | 0.02 |
| Small metro | 73,421 | 198 | 0.27 | 0.69 (0.58–0.82) | <0.01 |
| Micropolitan | 48,495 | 123 | 0.25 | 0.60 (0.49–0.74) | <0.01 |
| Noncore | 7,365 | 14 | 0.19 | 0.46 (0.27–0.79) | 0.01 |
| Unknown | 82,492 | 231 | 0.28 | 0.72 (0.60–0.86) | <0.01 |
| **Year of Index Prescription** | | | | | |
| 2015 | 201,359 | 711 | 0.35 | ref | ref |
| 2016 | 176,885 | 480 | 0.27 | 1.07 (0.94–1.20) | 0.31 |
| 2017 | 139,342 | 289 | 0.21 | 1.27 (1.10–1.47) | <0.01 |
| **Index Prescription MME** | | | | | |
| MME < = 75 | 151,909 | 440 | 0.29 | ref | ref |
| MME 76–100 | 108,737 | 248 | 0.23 | 0.90 (0.77–1.06) | 0.21 |
| MME 101–200 | 127,599 | 383 | 0.30 | 0.95 (0.81–1.11) | 0.53 |
| MME 201–300 | 85,019 | 226 | 0.27 | 0.83 (0.68–1.01) | 0.06 |
| MME >300 | 44,322 | 183 | 0.41 | 1.04 (0.82–1.31) | 0.78 |
| **Index Prescription Days Supply** | | | | | |

*(Continued)*

**Table 3.** (Continued)

|  | No Overdose | Overdose | Overdose Row % | Adjusted Hazard Ratio (95% Confidence Interval) | p value |
|---|---|---|---|---|---|
| < = 3 Days | 291,199 | 832 | 0.28 | ref | ref |
| 4–6 Days | 150,693 | 339 | 0.22 | 0.80 (0.70–0.92) | <0.01 |
| 7+ Days | 75,694 | 309 | 0.41 | 1.35 (1.14–1.59) | <0.01 |

SA, short-acting; MME, morphine milligram equivalents.

(n = 2,903) developed chronic use within the first year, compared to 2.4% (n = 1,822) among people filling an oxycodone-acetaminophen index prescription and 2.3% (n = 7,279) among people filling a hydrocodone-acetaminophen index prescription. Consistent with our previous modeling results, when dosage of the index prescription is not considered in the model, both oxycodone-acetaminophen (aOR = 1.22, 95% CI 1.15–1.30) and oxycodone monotherapy (aOR = 1.69, 95% CI 1.60–1.78) were associated with increased odds of first-year chronic use relative to hydrocodone-acetaminophen. However, when controlling for both patient and index prescription characteristics, patients who received oxycodone-acetaminophen were significantly *less likely* to experience first-year chronic use compared to those receiving hydrocodone-acetaminophen therapy (aOR 0.88, 95% CI 0.82–0.94). The likelihood of chronic use for those receiving oxycodone monotherapy was not significantly different from those receiving hydrocodone-acetaminophen (aOR 1.06, 95% CI 1.00–1.13) (S4 Table).

The total number of opioid overdoses in this subset of the sample was 1,346 (0.28%). Among individuals filling an oxycodone monotherapy index prescription, 0.44% experienced an opioid overdose in the study period, compared to 0.27% of those who received oxycodone-acetaminophen and 0.24% of those who received hydrocodone-acetaminophen. Relative to those receiving hydrocodone-acetaminophen, patients who received oxycodone monotherapy (aHR = 2.18, 95% CI 1.86–2.57) were at the greatest risk for opioid overdose, followed by those receiving oxycodone-acetaminophen combination therapy (aHR = 1.26, 95% CI 1.06–1.50), (S5 Table).

## Discussion

Using a comprehensive database, and after adjusting for multiple patient- and prescription-level factors, our study found that the risk of first-year chronic use after an initial opioid prescription was lower with oxycodone compared to hydrocodone (aOR 0.95, 95% CI 0.91–1.0) but the risk of opioid overdose was markedly higher following oxycodone (aHR 1.65, 95% CI 1.45–1.87). This finding should be notable for providers who write these prescriptions. The rates of chronic use for hydrocodone and oxycodone (2.53–3.33% respectively) and overdose (0.24–0.38% respectively) have significant implications on the lives of the patients affected by them. These findings vary from prior literature which used limited datasets such as only PDMP or only commercial insured data [5,13], and likely reflect this study's ability to capture prescriptions regardless of payer, control for prescription characteristics like MME, and control for patient level-factors including age, gender and comorbidities. The linkage with hospital discharge data allows an unprecedented description of overdose incidence after an initial prescription at the level of an entire state as compared to prior work that was limited only to certain populations.

An unexpected finding was the role of opioids in combination with acetaminophen. Whereas short-acting hydrocodone is always combined with other non-opioid active ingredients, prescribers have a choice when prescribing short-acting oxycodone. Prescribing an

opioid medication in combination form allows clinicians to provide convenient multi-modal therapy, whereas prescribing the active ingredients separately allows clinicians to maximize the dose of the non-opioid first before adding an opioid for acute pain. For the outcome of chronic use, combination oxycodone-acetaminophen was associated with lower odds of chronic use (aOR 0.9 (95% CI 0.8–0.9)), whereas the outcome with oxycodone monotherapy was similar to hydrocodone/acetaminophen (aOR 1.1 (95% CI (1.0–1.1)). Conversely, both forms of oxycodone were associated with a higher risk of overdose when compared with hydrocodone-acetaminophen, but oxycodone monotherapy was associated with the highest risk of overdose (aHR 2.2 (95% CI 1.9–2.6)). Because we are unable to detect if patients on monotherapy concurrently used over-the-counter acetaminophen, we cannot describe the protective effect with certainty, nor can we ascertain if the opioid prescription and/or formulation was based on perceived risk by the prescriber or patient request. We were also unable to detect if acetaminophen overdose was higher when combination products were prescribed.

For our primary outcomes, socioeconomic and demographic factors did influence results. In the adjusted analysis, older age, Black race, Medicaid alone or in dual enrollment with Medicare Advantage and living outside urban areas were associated with chronic use. These factors may be important to prescribers who begin treatment for painful conditions that may become chronic, such as back pain. For overdose risk, however, another pattern emerged. Compared to the reference group of age 18–24, only the most elderly group (75+) was more likely to experience overdose, highlighting the elevated risk of injury or death in this population [39]. Men were more likely to experience overdose than women and White patients were more likely to overdose compared with Hispanic and Asian-Pacific Islander individuals. Contrasting with chronic use, those in metro areas were more likely to experience overdose.

We identified an interesting pattern for the MME of the index prescription. Regardless of the agent, higher MME for the index prescription was strongly associated with development of chronic use, a finding consistent with other studies [13,40]. An index prescription MME of >300 (equivalent to forty 5 mg tablets of oxycodone or hydrocodone) had an aOR for chronic use of 3.9 (95% CI 3.6–4.2) compared to a prescription for < = 75 MME. When evaluating overdose, however, the adjusted analysis showed no difference in odds of overdose based on the MME of the initial prescription. Days' supply was associated with a higher risk only for prescriptions ≥7 days in duration (aHR 1.35 (95% CI 1.14–1.59)), although that variable may be unreliable given that it is calculated at the pharmacy level and may not be accurate for as-needed prescriptions. Although overdose was a rare event, the stronger association of oxycodone with overdose may indicate that the choice of agent is more important than the quantity prescribed, consistent with prior findings that the "likeability" of oxycodone may increase its harm potential [12]. Our finding that patients prescribed oxycodone were more likely to have an additional prescription beyond the first one further supports this hypothesis, although further work is also needed to determine if prescribers vary their prescription choice based on patient factors or indication. Our secondary analysis including a random prescriber effect for those with at least 200 patients prescribed an opioid yielded similar results, indicating that prescriber characteristics were unlikely to influence the findings.

## Limitations

Data analysis was retrospective and based upon a dataset used for administrative purposes; clinical details that might affect prescribing decisions were unavailable, including pain severity or indications for prescribing. Probabilistic matching in the merged dataset may have failed in some cases. Black patients comprised about 3% of the cohort and about half of the race/ethnicity data had to be imputed, rendering findings about race and ethnicity potentially less

accurate. Although we controlled for many sociodemographic factors, because of data use limitations we could not control for indication of the first prescription, such as major surgery vs. injury vs. a chronic painful illness (e.g. cancer). Therefore, it is possible that the results may differ if specific indications, including severity and chronicity of pain, were considered independently. Due to data availability, we included the Medicare Advantage population but did not include traditional Medicare fee-for-service insurance, meaning that the findings may not apply to the entire elderly population. ICD-10 codes do not perfectly delineate between overdoses from prescribed opioids vs. illicit opioids, so we could not define which type of opioid caused the overdose outcome. After the initial prescription, 41.6% of individuals received at least a second prescription, and 37.8% switched to another opioid, so the outcomes may not be directly related to the initial opioid dispensed. The chronic use analysis did not exclude people who died within a year of the initial dispensation, but our sensitivity analysis revealed that including them did not substantially change the results. Oregon's PDMP reports opioid prescriptions which were dispensed, but not if the patient consumed some or all of the prescription. Likewise, the PDMP does not capture prescriber specialty, so it was not possible to evaluate prescriber-related factors, which may have influenced the results. We included a random prescriber effect sub-analysis to partially explore this limitation and saw no differences in results. Finally, this study used data from Oregon and the findings may not apply to other settings.

In conclusion, among patients who received an index prescription of hydrocodone or oxycodone, hydrocodone is slightly more associated with subsequent chronic use when compared with oxycodone, but oxycodone is much more likely to be associated with future overdose. Opioid formulations in combination with acetaminophen may be relatively protective against these adverse outcomes. Additionally, certain patient characteristics that are more likely to be associated with both chronic use and overdose should be considered by prescribers. Given the risks, opioids should be prescribed to previously naïve patients only when felt it is absolutely necessary. When that decision is made, hydrocodone may be the favorable agent.

## Supporting information

**S1 Fig. Selection and inclusion of prescriptions and patients in the study.**
(TIF)

**S1 Table. ICD-9 or 10 diagnosis codes used to identify opioid-related overdose/poisoning for ED visits and inpatient hospitalization.**
(DOCX)

**S2 Table. Sensitivity analysis: Patient and index prescription characteristics associated with first-year chronic use, using an alternative chronic use definition.**
(DOCX)

**S3 Table. Multilevel cox regression sub-analysis: Patient and index prescription characteristics associated with opioid overdose.**
(DOCX)

**S4 Table. Combination drug sub-analysis: Patient and index prescription characteristics associated with first-year chronic use.**
(DOCX)

**S5 Table. Combination drug sub-analysis: Patient and index prescription characteristics associated with opioid overdose.**
(DOCX)

## Acknowledgments

The authors wish to thank Dagan Wright, Benjamin Chan, Dancia Hall, Craig New, Steven Ranzoni, Josh Van Otterloo, Peter Geissert, and Laura Chisolm at the Oregon Health Authority for their ongoing partnership, support, and insight.

## Author Contributions

**Conceptualization:** Scott G. Weiner, Sanae El Ibrahimi, Sara E. Hallvik, Christi Hildebran, Roger D. Weiss.

**Data curation:** Scott G. Weiner, Sanae El Ibrahimi, Sara E. Hallvik.

**Formal analysis:** Michelle A. Hendricks, Sanae El Ibrahimi, Grant A. Ritter.

**Funding acquisition:** Scott G. Weiner, Grant A. Ritter, Sara E. Hallvik, Christi Hildebran, Roger D. Weiss, Edward W. Boyer, Peter W. Kreiner.

**Investigation:** Scott G. Weiner, Michelle A. Hendricks, Sara E. Hallvik.

**Methodology:** Scott G. Weiner, Michelle A. Hendricks, Sanae El Ibrahimi, Grant A. Ritter, Sara E. Hallvik, Christi Hildebran, Peter W. Kreiner, Michael A. Fischer.

**Project administration:** Scott G. Weiner, Diana P. Flores.

**Supervision:** Scott G. Weiner, Christi Hildebran, Michael A. Fischer.

**Writing – original draft:** Scott G. Weiner, Michelle A. Hendricks, Sanae El Ibrahimi, Grant A. Ritter.

**Writing – review & editing:** Scott G. Weiner, Michelle A. Hendricks, Sanae El Ibrahimi, Grant A. Ritter, Sara E. Hallvik, Christi Hildebran, Roger D. Weiss, Edward W. Boyer, Diana P. Flores, Lewis S. Nelson, Peter W. Kreiner, Michael A. Fischer.

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
