## [Decision Letter · Decision Letter 0]

14 Dec 2021

PONE-D-21-33002Opioid-Related Overdose and Chronic Use Following an Initial Prescription of Hydrocodone versus OxycodonePLOS ONE

Dear Dr. Weiner,

Thank you for submitting your manuscript to PLOS ONE. After careful consideration, we feel that it has merit but does not fully meet PLOS ONE’s publication criteria as it currently stands. Therefore, we invite you to submit a revised version of the manuscript that addresses the points raised during the review process.

 Specifically, we would like you to consider several suggestions for analysis that the reviewers have outlined including:1. Justifying your chronic use definition and considering applying a more commonly applied definition as a sensitivity analysis.2. Speaking more to the potential impact of having no indication data in your data. If this data is available, I would strongly urge the authors to incorporate it into the analysis in some way.3. Consider a stratified analysis where you consider combination products separately from single-agent opioids.4. Incorporate prescriber-related factors as suggested by Reviewer 2. Furthermore, in revising the manuscript, we suggest the authors also consider the following:1. Refer to recommendations on language to reduce stigma and apply this to the revised text (e.g. avoiding use of the word 'abuse'. See https://www.drugabuse.gov/nidamed-medical-health-professionals/health-professions-education/words-matter-terms-to-use-avoid-when-talking-about-addiction for example)2. Add a limitation related to the fact that you don't know if people actually took the medication dispensed.3. Temper comments such as " In an era when opioid overdose deaths remain alarmingly high" - recent reporting from the CDC demonstrates how the illicit drug supply (ie. synthetic opioids like fentanyl) is now the predominant driver of the overdose crisis. As such, this strong wording should be softened.

We look forward to receiving your revised manuscript.

Kind regards,

Tara Gomes

Academic Editor

PLOS ONE

2. Please provide additional details regarding participant consent. If you are reporting a retrospective study of medical records, archived samples or third party data, please ensure that you have discussed whether all data were fully anonymized before you accessed them and/or whether the IRB or ethics committee waived the requirement for informed consent. If patients provided informed written consent to have data from their medical records used in research, please include this information.

“This work was funded by the National Institutes of Health, NIH 5-R01-DA044167 and 5-R01-HS026753.”

We note that you have provided information within the Acknowledgements Section. Please note that funding information should not appear in the Acknowledgments section or other areas of your manuscript. We will only publish funding information present in the Funding Statement section of the online submission form.

 “National Institute on Drug Abuse 5-R01-DA044167 (SGW, MAH, SEI, GAR, SEH, CH, RDW, EWB, DF, PWK) and Agency for Healthcare Research and Quality 5-R01-HS026753 (SGW). The funders had no role in study design, data collection and analysis, decision to publish, or preparation of the manuscript.”

Reviewers' comments:

Reviewer's Responses to Questions

**Comments to the Author**

1. Is the manuscript technically sound, and do the data support the conclusions?

Reviewer #1: Yes

Reviewer #2: Yes

2. Has the statistical analysis been performed appropriately and rigorously? 

Reviewer #1: Yes

Reviewer #2: Yes

3. Have the authors made all data underlying the findings in their manuscript fully available?

Reviewer #1: No

Reviewer #2: No

4. Is the manuscript presented in an intelligible fashion and written in standard English?

Reviewer #1: Yes

Reviewer #2: Yes

5. Review Comments to the Author

Reviewer #1: The authors conducted a study assessing predictors of chronic use and opioid-related overdose after an index prescription in a large linked retrospective cohort. While several other studies on this topic have been conducted, the study specifically focuses on the formulation dispensed in index dispensation (hydrocodone vs. oxycodone - two of the most commonly used opioids in the US). The large analysis (N=519,066) allowed for investigation into what are relatively rare outcomes (chronic use, opioid-related overdose). The analysis found that oxycodone is associated with a reduced risk of chronic use (even though the crude rate was higher for oxycodone), but increased risk of overdose. The study is well-conducted, although I have a couple of major comments related to adjustment of confounders in the model, need for further exploration into the change in direction of association for oxycodone and chronic use, and the chronic use definition. Also, a I think a key limitations is the lack of adjustment for indication, type of pain (e.g., acute, chronic), etc., which is touched upon briefly a couple times in Discussion but could be expanded upon.

1) Major comments

- Adjustment for dose of index dispensation is critical in the multivariable models, and adjustment for this confounder may be responsible for the change in direction of association between oxycodone and chronic use between the crude and adjusted analyses. Also, the finding of an association between increasing dose and chronic use, but not overdose risk, is one of the more interesting findings in the analysis. However, for the MME of the index dispensation, would it not make more sense to calculate and adjust for the average daily dose in MME across the index dispensation (cumulative MME divided by days supply) rather than the cumulative MME, particularly given that the authors also adjust for days’ supply in the model? I would imagine the cumulative MME and days’ supply would be highly correlated/collinear. Also, average daily dose may be a more meaningful value for physicians.

- As mentioned above, the crude chronic use rate was higher with Oxycodone but this relationship inverted after adjustment for other variables. I think it is important to investigate what confounder was responsible for changing the direction of association after adjustment. I am guessing that it is the dose but this would be good to fully unravel.

- Although touched upon briefly in the Discussion a couple times, it would be good to elaborate more on how the inability to adjust for severity of pain, indication, and chronic vs. acute pain may affect the analysis and the findings. I think this is a key limitation, as other studies assessing predictors of long-term use are able to adjust for indication or often focus on a specific acute event (e.g., surgery).

- Definitions for chronic/long-term use vary widely in literature and the chronic use definition in this analysis differs from one of the more commonly used definitions (see Von Korff et al - DOI: 10.1097/AJP.0b013e318169d03b). Please also see a review of long-term definitions and limitations of definitions used (https://onlinelibrary.wiley.com/doi/abs/10.1002/pds.4929). While this doesn't mean that the definition used in this paper is not valid, it might be good to address and discuss some of the limitations with the definition used. In particular, for people meeting the chronic use definition, it would be useful to know the average total days supply in the 365 days after index prescription. If my understanding is correct, it would be possible for some people to have only a very small number of days supply in these 365 days but still meet the chronic use definition (e.g., 6 dispensations with only 1 days drug supply in each, as an extreme example). I like the inclusion of the secondary outcome of cumulative MME because it overcomes some of the limitations of the chronic use definition.

2) Minor comments

Introduction

- The Introduction might want to include a greater focus on some of the negative consequences of long-term chronic opioid use (e.g., dependence, overdose, transitions to illicit drugs), given that this is one of the outcomes of the study. Transitions to illicit drugs is particularly important to mention given that the opioid overdose outcome used in this analysis is based on ICD codes that do not differentiate between prescription vs. illicit opioids. Also, since reducing overdose-related deaths is a key public health goal, it would also be good to discuss trends in opioid-related overdoses in Oregon within the Introduction (prescription vs. illicit, if possible).

- The Shah et al reference (#25) also compares oxycodone and hydrocodone (see paper's Supplementary material), and might want to be referenced in the Introduction.

- Lines 117-118 - Similar efficacy for acute or chronic pain, or both? It seems that the cited studies are mostly for acute pain.

- Lines 127-128 – It would be good to add 95% CIs for these odds ratios to provide insight into whether there is a statistically significant difference between the two.

- It might be worth discussing in Introduction or Discussion why some studies suggest higher risk of risk of chronic use with Oxycodone while others do not. Further, given that reference #5 was also performed by the study team, the Introduction should elaborate on how the current analysis builds upon this previous analysis.

Methods

- For the International audience, it would be good to have some additional information on the different health plans (Medicaid vs. medicare vs. commercial, etc), such as who is eligible etc. This is particularly important given the strong associations between insurance plan and the outcomes in this analysis.

- Lines 160-161 - Please provide rationale for why long-acting formulations were excluded, as it is likely that most of the studies cited in the Introduction (e.g., number of prescriptions dispensed in 2017, efficacy for pain, development of chronic use) included long-acting formulations?

- Opioid overdose definition – please reference other literature that has also used this definition. It would be good to mention here and in the Limitations that these codes capture overdoses from both illicit and prescription opioids.

- With regards to switching, is it possible hat an individual may have been dispensed two different types of opioid on the same day (oxycodone and another), and therefore they are not identifying an actual switch?

- Did the analysis with chronic use as the outcome exclude people who died within 365 days of the initial dispensation? If not, the limitations of not doing so should be noted (not all individuals had same follow-up to be classified as a chronic user), particularly given that people dispensed oxycodone were more likely to die from overdose (and potentially other causes).

Discussion

- Lines 300-301 – it would be good to add 95% CIs here and/or note that the findings were statistically significant.

In first paragraph of discussion, can compare to existing literature (references 5, 13 and 14, as noted in Introduction) and discuss why contrasting findings exist and how this study fills a gap in literature.

- Lines 306-307 – the paragraph starts by saying there are many potential explanations for these findings and then states that “the first” is related to combination formulations. However, there does not seem to be a second or third explanation within this paragraph or elsewhere in Discussion.

- Lines 311-314 – Can the authors comment on whether this explanation (ie. that combination formulations allow for lower dose thereby reducing overdose risk) would not be accounted for by adjusting for MME in the analysis?

- The authors suggest that combination formulations may be protective against overdose because it limits the dose of the opioid, but then also say that “prescribing the active ingredients separately allows clinicians to maximize the dose of the non-opioid before adding an opioid”. These statements seem to suggest that both strategies (ie. combination formulations and prescribing medications separately) could serve to limit the opioid dose.

- Somewhere in discussion, it would be good to compare the rate of transition to chronic use in their analysis (2.8%) to other studies assessing this transition rate.

- Lines 330-331 say that MME and days supply were not associated with overdose, but this isn’t true. Table 3 shows that initial days supply of 7+ is associated with higher risk of overdose.

- It would be good to note in the Discussion/Limitations that there was a high rate of switching and therefore chronic use or overdose might not be directly related to the initial opioid dispensed.

Reviewer #2: The authors present a retrospective cohort analysis of patients residing in Oregon, USA, who initiated hydrocodone or oxycodone as their first opioid prescription. The analysis uses novel linkages of databases including the Oregon prescription drug monitoring program (PDMP) and Oregon All Payer All Claims (APAC) databases. Primary outcomes included chronic opioid use and opioid overdose. The current analysis is rigorous, and the manuscript is well-written; however, I believe analyses related to health services factors and combination products could enhance the paper prior to publication.

Major considerations

1. My major concern is that the paper focuses on patient outcomes for those dispensed hydrocodone versus oxycodone, yet some of the outcomes are likely driven by unaccounted health services factors. Mainly, prescriber is not considered in the analysis. The choice of opioid is likely largely dependent on the prescriber. Prescriber preference is a well-established factor for exposures when two in-class drugs are shown to have clinical equipoise; importantly, prescriber may also be related to the clinical outcomes studied (e.g., patients seeing an oncologist are likely to become chronic opioid users vs. those prescribed a short opioid course after dental surgery). Based on my understanding, the NPI is a required entry the PDMP and should be available in the dataset. Is this the case?

a. If this information is available, I recommend a secondary analysis that examines the outcomes using the same multivariable models, but adds prescriber as a clustering variable (so, a multi-level model with NPI as a clustering variable).

b. At minimum, the authors should recognize lack of prescriber information as a limitation in their Discussion section and hypothesize how accounting for this information might affect results.

2. The authors briefly mention the potential that the use of combination agents for hydrocodone may in part explain results. To strengthen conclusions, I suggest to describe the proportion of initial prescriptions that were for a combination product, and, ideally, conduct a secondary analysis examining outcomes stratified by combination vs. single-agent product for the initial prescription (understanding that over the counter analgesic use will not be captured). The current analysis, which uses National Drug Codes, should be able to accommodate this.

Minor considerations

1. Given the large sample size, P-values are not particularly useful to examine differences in patient characteristics between exposure groups. Instead, I suggest the authors use standardized mean differences (Austin 2009 Stat Med [PMCID: PMC3472075] is helpful).

2. Please mention the dates of the study within the abstract for readers.

6. PLOS authors have the option to publish the peer review history of their article (what does this mean?). If published, this will include your full peer review and any attached files.

Reviewer #1: No

Reviewer #2: **Yes: **Kaleen Hayes

---

## [Author Response · Author response to Decision Letter 0]

9 Feb 2022

PONE-D-21-33002

Opioid-Related Overdose and Chronic Use Following an Initial Prescription of Hydrocodone versus Oxycodone

PLOS ONE

Dear Dr. Weiner,

Thank you for submitting your manuscript to PLOS ONE. After careful consideration, we feel that it has merit but does not fully meet PLOS ONE’s publication criteria as it currently stands. Therefore, we invite you to submit a revised version of the manuscript that addresses the points raised during the review process.

Response: Thank you. We are pleased to provide this revised version which addresses the majority of the Reviewers’ requests.

Specifically, we would like you to consider several suggestions for analysis that the reviewers have outlined including:

1. Justifying your chronic use definition and considering applying a more commonly applied definition as a sensitivity analysis.

Response: We conducted a sensitivity analysis to examine findings from the alternative definition (derived from Van Korff et al. as suggested by the reviewer below) and added it to the manuscript. The alternative definition was slightly more inclusive than our definition, but there was significant overlap between the definitions – 96% of those who met our study definition also met the alternative definition, while 85.3% of those who met the alternative definition also met our study definition. 

2. Speaking more to the potential impact of having no indication data in your data. If this data is available, I would strongly urge the authors to incorporate it into the analysis in some way.

Response: Unfortunately, due to data use limitations, we were not permitted to link diagnosis to opioid prescriptions in the PDMP. Therefore, we do not have indication data. We have stated this fact, and its implications, more clearly in the Limitations.

3. Consider a stratified analysis where you consider combination products separately from single-agent opioids.

Response: We had also wondered about this question and had prepared a separate manuscript addressing this request. However, we have decided to merge that analysis into this paper to address this suggestion. The results of subgroup analyses that address this issue have been added to the manuscript. The results are interesting, and show that individuals who received oxycodone in either form (either alone or in combination with acetaminophen) were more likely to experience overdose than individuals receiving hydrocodone-acetaminophen, and those receiving oxycodone monotherapy were at greatest risk of overdose. However, for chronic use, individuals receiving oxycodone-acetaminophen were significantly less likely to experience chronic use than those receiving hydrocodone-acetaminophen or oxycodone alone. 

4. Incorporate prescriber-related factors as suggested by Reviewer 2.

Response: We were able to update the chronic use models to include a random effect for prescriber that will help adjust the model for unmeasured prescriber factors that may affect prescribing and outcomes. Due to the memory limitations of our software, were not able to conduct a multilevel Cox regression on the entire sample on the overdose outcome. We were, however, able to run the multilevel Cox regression on a subset of prescribers with at least 200 patients who filled their opioid prescriptions. The results we obtained from this model were very similar to the results from the Cox regression without the prescriber random effect. We added this sub-analysis to the manuscript and a table showing the results is in the supplemental materials. 

Note, that Oregon’s PDMP does not capture prescriber characteristics, such as specialty. The database does include DEA/NPI numbers and it would be theoretically possible to look these up on an individual level. Unfortunately, due to data use protections, the final linked database did not include that information and we are unable to go back and obtain it. 

Furthermore, in revising the manuscript, we suggest the authors also consider the following:

1. Refer to recommendations on language to reduce stigma and apply this to the revised text (e.g. avoiding use of the word 'abuse'. See https://www.drugabuse.gov/nidamed-medical-health-professionals/health-professions-education/words-matter-terms-to-use-avoid-when-talking-about-addiction for example)

Response: We are very attuned to the issue of stigma. The only places we used the word “abuse” were in reference to medications and not individuals (i.e. “abuse potential” or “abuse liability”), which are commonly used in the medical literature. However, we have changed this to “misuse” as it captures the same concept and hopefully will not be perceived as stigmatizing.

2. Add a limitation related to the fact that you don't know if people actually took the medication dispensed.

Response: Done.

3. Temper comments such as " In an era when opioid overdose deaths remain alarmingly high" - recent reporting from the CDC demonstrates how the illicit drug supply (ie. synthetic opioids like fentanyl) is now the predominant driver of the overdose crisis. As such, this strong wording should be softened.

Response: Done. We have removed this phrase.

We look forward to receiving your revised manuscript.

Kind regards,

Tara Gomes

Academic Editor

PLOS ONE

Response: Done

Response: Done

2. Please provide additional details regarding participant consent. If you are reporting a retrospective study of medical records, archived samples or third party data, please ensure that you have discussed whether all data were fully anonymized before you accessed them and/or whether the IRB or ethics committee waived the requirement for informed consent. If patients provided informed written consent to have data from their medical records used in research, please include this information.

Response: Done

Response: Done

“This work was funded by the National Institutes of Health, NIH 5-R01-DA044167 and 5-R01-HS026753.”

We note that you have provided information within the Acknowledgements Section. Please note that funding information should not appear in the Acknowledgments section or other areas of your manuscript. We will only publish funding information present in the Funding Statement section of the online submission form.

Response: Done

 “National Institute on Drug Abuse 5-R01-DA044167 (SGW, MAH, SEI, GAR, SEH, CH, RDW, EWB, DF, PWK) and Agency for Healthcare Research and Quality 5-R01-HS026753 (SGW). The funders had no role in study design, data collection and analysis, decision to publish, or preparation of the manuscript.”

Response: Thank you. We have included this in the cover letter. There is one small edit (addition of a middle initial):

“National Institute on Drug Abuse 5-R01-DA044167 (SGW, MAH, SEI, GAR, SEH, CH, RDW, EWB, DPF, PWK) and Agency for Healthcare Research and Quality 5-R01-HS026753 (SGW). The funders had no role in study design, data collection and analysis, decision to publish, or preparation of the manuscript.”

Reviewers' comments:

Reviewer's Responses to Questions

Comments to the Author

1. Is the manuscript technically sound, and do the data support the conclusions?

Reviewer #1: Yes

Reviewer #2: Yes

2. Has the statistical analysis been performed appropriately and rigorously?

Reviewer #1: Yes

Reviewer #2: Yes

3. Have the authors made all data underlying the findings in their manuscript fully available?

Reviewer #1: No

Reviewer #2: No

Response: The terms of the data use agreement prohibit public sharing of the source data used in this study but may be obtained via request to the public health entities in Oregon that oversee the individual source datasets. We have indicated this in the text.

4. Is the manuscript presented in an intelligible fashion and written in standard English?

Reviewer #1: Yes

Reviewer #2: Yes

5. Review Comments to the Author

Reviewer #1: The authors conducted a study assessing predictors of chronic use and opioid-related overdose after an index prescription in a large linked retrospective cohort. While several other studies on this topic have been conducted, the study specifically focuses on the formulation dispensed in index dispensation (hydrocodone vs. oxycodone - two of the most commonly used opioids in the US). The large analysis (N=519,066) allowed for investigation into what are relatively rare outcomes (chronic use, opioid-related overdose). The analysis found that oxycodone is associated with a reduced risk of chronic use (even though the crude rate was higher for oxycodone), but increased risk of overdose. The study is well-conducted, although I have a couple of major comments related to adjustment of confounders in the model, need for further exploration into the change in direction of association for oxycodone and chronic use, and the chronic use definition. Also, a I think a key limitations is the lack of adjustment for indication, type of pain (e.g., acute, chronic), etc., which is touched upon briefly a couple times in Discussion but could be expanded upon.

1) Major comments

- Adjustment for dose of index dispensation is critical in the multivariable models, and adjustment for this confounder may be responsible for the change in direction of association between oxycodone and chronic use between the crude and adjusted analyses. Also, the finding of an association between increasing dose and chronic use, but not overdose risk, is one of the more interesting findings in the analysis. However, for the MME of the index dispensation, would it not make more sense to calculate and adjust for the average daily dose in MME across the index dispensation (cumulative MME divided by days supply) rather than the cumulative MME, particularly given that the authors also adjust for days’ supply in the model? I would imagine the cumulative MME and days’ supply would be highly correlated/collinear. Also, average daily dose may be a more meaningful value for physicians.

Response: Thank you. We agree with the interesting finding about increasing dose and chronic use (but not overdose risk). We were hesitant to include days’ supply in the model because it can be problematic in PDMP-based opioid-related research because, for acute opioid prescriptions, prescribers don’t write an intended number of days for “as needed” prescriptions. Days are calculated at the pharmacy or PDMP level based on if a patient were to take the maximum number of pills. As an example, a prescription for 2 tablets, every six hours as needed for severe pain, 24 tablets would be recorded as a 3 days’ supply even if the prescriber didn’t intend the patient to take them all in 3 days. That reality also discourages us from calculating cumulate MME/days’ supply. However, day’s supply and MME are not multi-collinear, so they are both included in the model.

- As mentioned above, the crude chronic use rate was higher with oxycodone but this relationship inverted after adjustment for other variables. I think it is important to investigate what confounder was responsible for changing the direction of association after adjustment. I am guessing that it is the dose but this would be good to fully unravel.

Response: We investigated this within the modeling and found that inclusion of MME in the model was responsible for the change in relationship after confounder adjustments. These findings have been added to the manuscript. 

- Although touched upon briefly in the Discussion a couple times, it would be good to elaborate more on how the inability to adjust for severity of pain, indication, and chronic vs. acute pain may affect the analysis and the findings. I think this is a key limitation, as other studies assessing predictors of long-term use are able to adjust for indication or often focus on a specific acute event (e.g., surgery).

Response: Done. We have added additional language to this Limitations section.

- Definitions for chronic/long-term use vary widely in literature and the chronic use definition in this analysis differs from one of the more commonly used definitions (see Von Korff et al - DOI: 10.1097/AJP.0b013e318169d03b). Please also see a review of long-term definitions and limitations of definitions used (https://onlinelibrary.wiley.com/doi/abs/10.1002/pds.4929). While this doesn't mean that the definition used in this paper is not valid, it might be good to address and discuss some of the limitations with the definition used. In particular, for people meeting the chronic use definition, it would be useful to know the average total days supply in the 365 days after index prescription. If my understanding is correct, it would be possible for some people to have only a very small number of days supply in these 365 days but still meet the chronic use definition (e.g., 6 dispensations with only 1 days drug supply in each, as an extreme example). I like the inclusion of the secondary outcome of cumulative MME because it overcomes some of the limitations of the chronic use definition.

Response: We conducted a sensitivity analysis to examine findings from the alternative definition (derived from Van Korff et al) and added it to the manuscript. We also added the median total days’ supply for individuals meeting each definition to the text. The alternative definition was slightly more inclusive than our definition, but there was significant overlap between the definitions – 96% of those who met our study definition also met the alternative definition, while 85.3% of those who met the alternative definition also met our study definition. 

2) Minor comments

Introduction

- The Introduction might want to include a greater focus on some of the negative consequences of long-term chronic opioid use (e.g., dependence, overdose, transitions to illicit drugs), given that this is one of the outcomes of the study. Transitions to illicit drugs is particularly important to mention given that the opioid overdose outcome used in this analysis is based on ICD codes that do not differentiate between prescription vs. illicit opioids. Also, since reducing overdose-related deaths is a key public health goal, it would also be good to discuss trends in opioid-related overdoses in Oregon within the Introduction (prescription vs. illicit, if possible).

Response: As suggested, we have added two additional paragraphs to the Introduction discussing the negative consequences of opioid use, transition to illicit opioids, and specifics about Oregon.

- The Shah et al reference (#25) also compares oxycodone and hydrocodone (see paper's Supplementary material), and might want to be referenced in the Introduction.

Response: Excellent suggestion – we were unaware of that supplementary material. We have now included this information in the Introduction.

- Lines 117-118 - Similar efficacy for acute or chronic pain, or both? It seems that the cited studies are mostly for acute pain.

Response: We split this sentence into two to make it clearer to the reader.

- Lines 127-128 – It would be good to add 95% CIs for these odds ratios to provide insight into whether there is a statistically significant difference between the two.

Response: Done. The CIs do not overlap.

- It might be worth discussing in Introduction or Discussion why some studies suggest higher risk of risk of chronic use with Oxycodone while others do not. Further, given that reference #5 was also performed by the study team, the Introduction should elaborate on how the current analysis builds upon this previous analysis.

Response: We have attempted to explain this concept better in this version of the manuscript. We believe prior studies have been subject to several limitations. The Ohio study was limited to PDMP data and did not control for any patient-level data. Other studies are limited to certain insurance datasets and do not include self-pay patients. We believe our study provides the most compelling evidence to date, and also includes the overdose outcome which has been incomplete in past work.

Methods

- For the International audience, it would be good to have some additional information on the different health plans (Medicaid vs. medicare vs. commercial, etc), such as who is eligible etc. This is particularly important given the strong associations between insurance plan and the outcomes in this analysis.

Response: Done. We have added a sentence briefly describing the different insurance types.

- Lines 160-161 - Please provide rationale for why long-acting formulations were excluded, as it is likely that most of the studies cited in the Introduction (e.g., number of prescriptions dispensed in 2017, efficacy for pain, development of chronic use) included long-acting formulations?

Response: This has been added. Long-acting formulations were excluded due to the small percentage of index prescriptions that were long-acting in our sample (e.g., 0.05% of hydrocodone or oxycodone index prescriptions in 2015). 

- Opioid overdose definition – please reference other literature that has also used this definition. It would be good to mention here and in the Limitations that these codes capture overdoses from both illicit and prescription opioids.

Response: We have added to the methods that we are capturing both illicit and prescription opioids in our definition of overdose. We have referenced SAMHSA’s guidance for defining opioid overdose. In the limitations, we now state: “ICD-10 codes do not perfectly delineate between overdoses from prescribed opioids vs. illicit opioids, so we could not define which type of opioid caused the overdose outcome.”

- With regards to switching, is it possible that an individual may have been dispensed two different types of opioid on the same day (oxycodone and another), and therefore they are not identifying an actual switch?

Response: This is not possible because, as stated in the Methods, individuals dispensed more than one opioid on their index prescription date were excluded from the analysis. 

- Did the analysis with chronic use as the outcome exclude people who died within 365 days of the initial dispensation? If not, the limitations of not doing so should be noted (not all individuals had same follow-up to be classified as a chronic user), particularly given that people dispensed oxycodone were more likely to die from overdose (and potentially other causes).

Response: Excellent point. No, we did not exclude people who died within 365 days of the initial dispensation, and have now listed this in the limitations as suggested.

Discussion

- Lines 300-301 – it would be good to add 95% CIs here and/or note that the findings were statistically significant.

Response: Done.

In first paragraph of discussion, can compare to existing literature (references 5, 13 and 14, as noted in Introduction) and discuss why contrasting findings exist and how this study fills a gap in literature.

Response: We are happy to make this suggested change, which highlights the novelty and rigor of our study.

- Lines 306-307 – the paragraph starts by saying there are many potential explanations for these findings and then states that “the first” is related to combination formulations. However, there does not seem to be a second or third explanation within this paragraph or elsewhere in Discussion.

Response: The subsequent reasons are in the following paragraphs. To make this clearer, we have modified the first sentence of the following paragraphs.

- Lines 311-314 – Can the authors comment on whether this explanation (ie. that combination formulations allow for lower dose thereby reducing overdose risk) would not be accounted for by adjusting for MME in the analysis?

Response: This question has been addressed by the new secondary analysis of combination vs. solo formulations of oxycodone.

- The authors suggest that combination formulations may be protective against overdose because it limits the dose of the opioid, but then also say that “prescribing the active ingredients separately allows clinicians to maximize the dose of the non-opioid before adding an opioid”. These statements seem to suggest that both strategies (ie. combination formulations and prescribing medications separately) could serve to limit the opioid dose.

Response: Here, we intend that a prescriber can fully separate out the agents. That is, they can maximize use of non-opioids, and then only add an opioid when the non-opioids are not working. With combination pills, the prescriber starts immediately with an opioid. Therefore, the strategy of separating the agents allows for no opioid use vs. some opioid use. We added the word “first” to this sentence which we believe makes the point easier to understand.

- Somewhere in discussion, it would be good to compare the rate of transition to chronic use in their analysis (2.8%) to other studies assessing this transition rate.

Response: Done. We have added text in both the Introduction and Discussion to address this suggestion.

- Lines 330-331 say that MME and days supply were not associated with overdose, but this isn’t true. Table 3 shows that initial days supply of 7+ is associated with higher risk of overdose.

Response: The Reviewer is correct, and we have made this change.

- It would be good to note in the Discussion/Limitations that there was a high rate of switching and therefore chronic use or overdose might not be directly related to the initial opioid dispensed.

Response: Excellent point. We have added this to the Limitations.

Reviewer #2: The authors present a retrospective cohort analysis of patients residing in Oregon, USA, who initiated hydrocodone or oxycodone as their first opioid prescription. The analysis uses novel linkages of databases including the Oregon prescription drug monitoring program (PDMP) and Oregon All Payer All Claims (APAC) databases. Primary outcomes included chronic opioid use and opioid overdose. The current analysis is rigorous, and the manuscript is well-written; however, I believe analyses related to health services factors and combination products could enhance the paper prior to publication.

Response: Thank you for these comments. We address the suggestions below.

Major considerations

1. My major concern is that the paper focuses on patient outcomes for those dispensed hydrocodone versus oxycodone, yet some of the outcomes are likely driven by unaccounted health services factors. Mainly, prescriber is not considered in the analysis. The choice of opioid is likely largely dependent on the prescriber. Prescriber preference is a well-established factor for exposures when two in-class drugs are shown to have clinical equipoise; importantly, prescriber may also be related to the clinical outcomes studied (e.g., patients seeing an oncologist are likely to become chronic opioid users vs. those prescribed a short opioid course after dental surgery). Based on my understanding, the NPI is a required entry the PDMP and should be available in the dataset. Is this the case?

a. If this information is available, I recommend a secondary analysis that examines the outcomes using the same multivariable models, but adds prescriber as a clustering variable (so, a multi-level model with NPI as a clustering variable).

b. At minimum, the authors should recognize lack of prescriber information as a limitation in their Discussion section and hypothesize how accounting for this information might affect results.

Response: See our comment above regarding this issue. We do not have much in the way of prescriber factors, but we were able to add a random effect for prescriber to most of our models. For the Cox regression, we had to run a model on a subset of the sample due to memory limitations, but results were similar to the original models that did not include a prescriber random effect. 

2. The authors briefly mention the potential that the use of combination agents for hydrocodone may in part explain results. To strengthen conclusions, I suggest to describe the proportion of initial prescriptions that were for a combination product, and, ideally, conduct a secondary analysis examining outcomes stratified by combination vs. single-agent product for the initial prescription (understanding that over the counter analgesic use will not be captured). The current analysis, which uses National Drug Codes, should be able to accommodate this.

Response: We have added an additional sub-analysis analysis that compares chronic use and overdose risk for combination products compared to monotherapy. 

Minor considerations

1. Given the large sample size, P-values are not particularly useful to examine differences in patient characteristics between exposure groups. Instead, I suggest the authors use standardized mean differences (Austin 2009 Stat Med [PMCID: PMC3472075] is helpful).

Response: This has been updated.

2. Please mention the dates of the study within the abstract for readers.

Response: Done.

6. PLOS authors have the option to publish the peer review history of their article (what does this mean?). If published, this will include your full peer review and any attached files.

Do you want your identity to be public for this peer review? For information about this choice, including consent withdrawal, please see our Privacy Policy.

Reviewer #1: No

Reviewer #2: Yes: Kaleen Hayes

---

## [Editor Report · Decision Letter 1]

11 Mar 2022

PONE-D-21-33002R1

Opioid-Related Overdose and Chronic Use Following an Initial Prescription of Hydrocodone versus Oxycodone

PLOS ONE

Dear Dr. Weiner,

Thank you for submitting your manuscript to PLOS ONE. After careful consideration, we feel that it has merit but does not fully meet PLOS ONE’s publication criteria as it currently stands. Therefore, we invite you to submit a revised version of the manuscript that addresses the points raised during the review process.

1. The changes to the introduction have strengthened the paper, however the rationale in your final paragraph which states "Given these hazards associated with opioid use, and because oxycodone and hydrocodone appear to be used interchangeably for acute pain..." doesn't follow from the paragraph above where you outline that rises in opioid-related deaths are being driven by the illicit supply. I would suggest changing this sentence to address the fact that current rising trends are not being driven by prescriptions, but that despite this, there is a need to ensure safe prescription opioid practices. 

2. You may want to consider including citations to the revised CDC guidelines that are now released for comment (https://www.regulations.gov/document/CDC-2022-0024-0002) as they provide more up-to-date guidance related to opioid use for CNCP.

3. When describing the alternative definition of chronic use, consider rephrasing for clarity as "...with either a total days’ supply greater than or equal to 120 or 10+ prescriptions filled." 

4. As one of the reviewers pointed out originally, there are limitations to the decision not to exclude those who died in the 365 days of follow-up as they had a different timeline to meet the chronic use criteria. While you added this as a limitation in the revision, sufficient details weren't provided to understand the impact of this limitation. Can you please provide further detail as to how deaths were accounted for in your chronic use definition (ie when defining periods uncovered by opioid prescriptions being <30 days, what if someone died 60 days before the 1-year follow-up period ended? Would that be counted as a 60 day period 'uncovered' or did you only consider the period of time when they were alive to identify gaps?). Can you also provide the N(%) of people who died during the first year by exposure group to allow an assessment of whether the prevalence of death differed by group (which could therefore bias the findings), or whether it was fairly rare/similar between groups (in which case the decision not to exclude these individuals likely is less problematic).

We look forward to receiving your revised manuscript.

Kind regards,

Tara Gomes

Academic Editor

PLOS ONE
---

## [Author Response · Author response to Decision Letter 1]

17 Mar 2022

Thank you for submitting your manuscript to PLOS ONE. After careful consideration, we feel that it has merit but does not fully meet PLOS ONE’s publication criteria as it currently stands. Therefore, we invite you to submit a revised version of the manuscript that addresses the points raised during the review process.

Response: Thank you for the additional review. We are happy to address these remaining concerns.

1. The changes to the introduction have strengthened the paper, however the rationale in your final paragraph which states "Given these hazards associated with opioid use, and because oxycodone and hydrocodone appear to be used interchangeably for acute pain..." doesn't follow from the paragraph above where you outline that rises in opioid-related deaths are being driven by the illicit supply. I would suggest changing this sentence to address the fact that current rising trends are not being driven by prescriptions, but that despite this, there is a need to ensure safe prescription opioid practices. 

Response: We have made the adjustment to this paragraph as suggested.

2. You may want to consider including citations to the revised CDC guidelines that are now released for comment (https://www.regulations.gov/document/CDC-2022-0024-0002) as they provide more up-to-date guidance related to opioid use for CNCP.

Response: This is a delicate balance as we don’t know when the new guidelines will come out nor how they will differ from the version which is open to public comment presently (until April 11). We therefore modified the sentence about the guidelines to read: “Due to these and other complications, avoidance of chronic opioid therapy for pain not related to malignancy is recommended by the Centers for Disease Control and Prevention’s 2016 opioid guidelines, although an updated version is currently forthcoming [19, 20].” Hopefully, this edit adequately addresses the suggestion.

3. When describing the alternative definition of chronic use, consider rephrasing for clarity as "...with either a total days’ supply greater than or equal to 120 or 10+ prescriptions filled."

Response: We have made the suggested edit.

4. As one of the reviewers pointed out originally, there are limitations to the decision not to exclude those who died in the 365 days of follow-up as they had a different timeline to meet the chronic use criteria. While you added this as a limitation in the revision, sufficient details weren't provided to understand the impact of this limitation. Can you please provide further detail as to how deaths were accounted for in your chronic use definition (ie when defining periods uncovered by opioid prescriptions being <30 days, what if someone died 60 days before the 1-year follow-up period ended? Would that be counted as a 60 day period 'uncovered' or did you only consider the period of time when they were alive to identify gaps?). Can you also provide the N(%) of people who died during the first year by exposure group to allow an assessment of whether the prevalence of death differed by group (which could therefore bias the findings), or whether it was fairly rare/similar between groups (in which case the decision not to exclude these individuals likely is less problematic).

Response: For people who died, we did not limit the follow-up period to the time they were alive. Among those receiving an index prescription of hydrocodone SA, 1.06% (n = 3,750) died during the follow-up year. Among those receiving oxycodone SA, 1.73% (n = 2,868) died during the follow-up year. We also reran the chronic use analysis after excluding individuals that died during the follow-up year. The OR for oxycodone SA was 0.94 (95% CI 0.89 – 0.98; reference = hydrocodone SA). Comparing this to the estimate for oxycodone SA in the original model that did not exclude individuals who died during the follow-up (OR 0.95, 95% CI 0.91-1.00) demonstrated very little change in the results. We have added these findings to the Limitations section.

---

## [Editor Report · Decision Letter 2]

21 Mar 2022

PONE-D-21-33002R2Opioid-Related Overdose and Chronic Use Following an Initial Prescription of Hydrocodone versus OxycodonePLOS ONE

Dear Dr. Weiner,

Thank you for submitting your manuscript to PLOS ONE. After careful consideration, we feel that it has merit but does not fully meet PLOS ONE’s publication criteria as it currently stands. Therefore, we invite you to submit a revised version of the manuscript that addresses the points raised during the review process.

I appreciate you adding in the sensitivity analysis in which you excluded people who died in the first year of follow-up. However, to follow journal style, please incorporate this analysis as a sensitivity analysis into the methods and results of the manuscript (this can be brief), and then simply speak to the consistency in findings between the primary and sensitivity analysis in the limitations section of the discussion.

We look forward to receiving your revised manuscript.

Kind regards,

Tara Gomes

Academic Editor

PLOS ONE
---

## [Author Response · Author response to Decision Letter 2]

21 Mar 2022

Thank you for the additional suggestion, which we are happy to address.

As requested, we have moved the sensitivity analysis from the Limitations to the Methods and Results, leaving one sentence in Limitations. Hopefully, the article now meets the journal's formatting requirements.

---

## [Editor Report · Decision Letter 3]

23 Mar 2022

Opioid-Related Overdose and Chronic Use Following an Initial Prescription of Hydrocodone versus Oxycodone

PONE-D-21-33002R3

Dear Dr. Weiner,

We’re pleased to inform you that your manuscript has been judged scientifically suitable for publication and will be formally accepted for publication once it meets all outstanding technical requirements.

Kind regards,

Tara Gomes

Academic Editor

PLOS ONE
---

## [Editor Report · Acceptance letter]

28 Mar 2022

PONE-D-21-33002R3 

Opioid-related overdose and chronic use following an initial prescription of hydrocodone versus oxycodone 

Dear Dr. Weiner:

I'm pleased to inform you that your manuscript has been deemed suitable for publication in PLOS ONE. Congratulations! Your manuscript is now with our production department. 

Kind regards, 

on behalf of

Dr. Tara Gomes 

Academic Editor

PLOS ONE